# Role of Oleic Acid in the Gut-Liver Axis: From Diet to the Regulation of Its Synthesis via Stearoyl-CoA Desaturase 1 (SCD1)

**DOI:** 10.3390/nu11102283

**Published:** 2019-09-24

**Authors:** Elena Piccinin, Marica Cariello, Stefania De Santis, Simon Ducheix, Carlo Sabbà, James M. Ntambi, Antonio Moschetta

**Affiliations:** 1Clinica Medica Cesare Frugoni, Department of Interdisciplinary Medicine, University of Bari Aldo Moro, 70124 Bari, Italy; elena.piccinin@uniba.it (E.P.); marica.cariello@uniba.it (M.C.); stefania.desantis@uniba.it (S.D.S.); carlo.sabba@uniba.it (C.S.); 2INBB, National Institute for Biostructures and Biosystems, 00136 Rome, Italy; 3Department of Pharmacy-Drug Science, University of Bari “Aldo Moro”, 70126 Bari, Italy; 4Institut du thorax, INSERM, CNRS, University of Nantes, 44007 Nantes, France; simon.ducheix@univ-nantes.fr; 5Departments of Biochemistry and of Nutritional Sciences, University of Wisconsin, Madison, WI 53706, USA; james.ntambi@wisc.edu; 6IRCCS Istituto Tumori Giovanni Paolo II, 70124 Bari, Italy

**Keywords:** olive oil, oleic acid, MUFA, stearoyl-CoA desaturase, liver, gut

## Abstract

The consumption of an olive oil rich diet has been associated with the diminished incidence of cardiovascular disease and cancer. Several studies have attributed these beneficial effects to oleic acid (C18 n-9), the predominant fatty acid principal component of olive oil. Oleic acid is not an essential fatty acid since it can be endogenously synthesized in humans. Stearoyl-CoA desaturase 1 (SCD1) is the enzyme responsible for oleic acid production and, more generally, for the synthesis of monounsaturated fatty acids (MUFA). The saturated to monounsaturated fatty acid ratio affects the regulation of cell growth and differentiation, and alteration in this ratio has been implicated in a variety of diseases, such as liver dysfunction and intestinal inflammation. In this review, we discuss our current understanding of the impact of gene-nutrient interactions in liver and gut diseases, by taking advantage of the role of SCD1 and its product oleic acid in the modulation of different hepatic and intestinal metabolic pathways.

## 1. Introduction

Dietary recommendations suggest that fat should represent 20% to 35% of energy intake and the type of fat ingested is a fundamental clue in the prevention of several diseases [1]. Indeed, fat overconsumption and fat quality have been linked to obesity, insulin resistance and the metabolic syndrome [2]. Dietary fats and oils are composed of triglycerides made of three fatty acids esterified on a glycerol backbone. As fatty acids are highly diverse, the classifications of fatty acids are based on their carbon chain length, the number and the position of the unsaturations (double bounds) present between two carbons. They are classified into different families: saturated FA (SFA), with no unsaturation such as palmitic or stearic acids, monounsaturated (MUFA)—if it contains one double bond—and polyunsaturated (PUFA)—if there are more than one. The palmitoleic acid (16:1) and the oleic acid (18:1) represent two important MUFA. The PUFAs can further be subdivided into the two renowned omega-6 (*n*-6) (linoleic acid, arachidonic acid) and omega-3 (*n*-3) acids (α-linolenic acid, eicosapentaenoic acid (EPA), and docosahexaenoic acid (DHA)) based on the distance from the first double bond to the methyl end, although PUFA *n*-7 and *n*-9 exist [3]. Both omega-6 and omega-3 are essential fatty acids and must be obtained exclusively from the dietary sources.

According to conventional dietary recommendations, a healthy diet should include a daily consumption of fats: an inadequate quantity of fat might increase the risk of conditions such as depression, psoriasis, Alzheimer’s disease and schizophrenia [4,5]. To date, several edible oils have been characterized for their health benefits, particularly due to their abundance in MUFA and/or PUFA. Extra-virgin olive oil (EVOO), sunflower oil (SFO), palm oil, and fish oil have been extensively studied for their contribution to disease prevention. However, other types of edible oils may play a role in preserving health status. For instance, due to their high concentration in MUFA and PUFA and a low quantity of SFA, mustard and canola oils display beneficial effects on lipid serum profile and the cardiovascular system [6,7].

EVOO represents the main source of fat in the Mediterranean diet [8]. It is constituted by a high content of MUFA (mostly oleic acid), a variable but significant amount of PUFA (ranging from 1.5% to 21%), and minor amounts of antioxidant micronutrients such as polyphenols, squalene, lignans, phenyl-ethyl alcohols and secoiridoids [9,10]. Oleic acid (18:1 n-9) represents 49% to 83% of total FA in olive oil [11] and its consumption has been related to improved pancreas and liver secretory activity as well as reduced gastric-duodenal ulcers risk [12]. Moreover, MUFAs are able to modify plasma lipids and lipoprotein composition and hence reduce inflammation, oxidative stress and coagulation and ameliorate glucose homeostasis and blood pressure [12]. This evidence has confirmed the beneficial properties of EVOO bioactive components that have led to the established health claims by FDA and EFSA. Both agencies stated that oleic acid contained in olive oil, together with the present polyphenols, contributes to the maintenance of normal blood cholesterol levels [13,14].

The SFO contains a large amount of PUFA (linoleic acid 60% to 70%), but MUFA (oleic acid), SFA (stearic acid) and tocopherol are present as well in adequate quantities [15,16]. This oil has traditionally been used as control in several studies [15,17]. It has been shown that SFO can modify the serum lipid profile by reducing total cholesterol and low-density lipoprotein (LDL) levels [18,19]. On the other hand, a diet rich in linoleic acid increases reactive oxygen species (ROS) production compared to a SFA-rich diet, thereby promoting lipid peroxidation [2].

Different from the previous two fatty acids, fish oil is a good source of the B12 and D vitamins and omega-3 FAs (EPA and DHA) [20,21]. The favourable health effects of fish oil were initially established by the remarkably low incidence of coronary artery disease within the Inuit community, despite consuming a high-fat diet [22]. These health effects have largely been attributed to EPA and DHA. Indeed, several studies have reported that diet supplementation with purified EPA and DHA for more than three years result in cardioprotective effects [23,24]. In the past years, several meta-analyses and systematic reviews have been published in order to evaluate the association between fish oil and protection from cardiovascular risk [25,26,27,28,29,30,31,32]. Most of the studies reported no significant association of omega-3 supplementation with fatal or nonfatal coronary heart disease or any major vascular events. Furthermore, a large cohort study recently demonstrated that diet supplemented with fish oil containing omega-3 failed to reduce the incidence of major cardiovascular events or cancer [33]. If omega-3 FA supplement containing EPA and DHA is approved from FDA, it is not the same for the other dietary supplements of fish oil. Different from the first, fish oil supplement can include other fats and cholesterols as well, which may negatively affect the health response [34].

Palm oil is constituted of 51% SFA (palmitic acid), 38% MUFA, 11% PUFA [35], carotenoids, lycopene and xanthophylls [36]. Rats fed with 12% palm oil for one-year display increased total serum cholesterol levels, but lower aortic accumulation of cholesteryl esters compared to SFO, soybean and butter most likely due to the high concentration of micronutrients and MUFA in this oil [18]. Moreover, palm oil may reduce oxidative stress associated to ischemia-reperfusion injury [37]. Further studies are needed to clarify the contribution of these edible oils on human health, evaluating the role played by different fatty acids.

It is becoming evidently clear that the various dietary oils differ not only in the FA profile, but in the content of non-saponifiable species as well. Studies on olive oil have regularly been more focused on these non-saponifiable components. On the contrary, the direct role of oleic acid has been neglected. Different from other FA, oleic acid is not an essential FA, since it can be de novo synthesized by stearoyl-CoA desaturase 1 (SCD1). Exploring SCD1, the master regulator of oleic acid synthesis, may therefore offer important insights on the role of the oleic acid largely present in olive oil. In this light, in vivo and in vitro studies in which the activity of SCD1 has been shut down represent a valid opportunity to evaluate the direct contribution of oleic acid supplementation with the diet (i.e. EVOO). To date, several clinical trials are currently conducted focusing on SCD1 (Table 1) aiming to understand how nutrients may interact with the organism, regulating energy and metabolic homeostasis as well as disease progression.

Lipids ingested with the diet are primarily digested and absorbed in the small intestine, followed by delivery to their sites of action in the body by crossing the liver. Thus, the gastrointestinal tract and the liver are at the nexus between a vast source of nutrients and the rest of the body and are connected both anatomically and functionally. The gut-liver axis is characterized by bidirectional traffic: nutrients and factors derived from gut lumen reach the liver through the portal circulation, while bile acids are released in the small intestine from the biliary duct.

In this review, we will particularly focus on the role of the major component of olive oil—the oleic acid. We will exploit the contribution of SCD1 and its product oleic acid in health and diseases, with particular attention to the gut-liver axis. Although it might appear controversial in some cases, it is evidently clear that oleic acid plays an essential part in the development and the homeostasis of our organism, and possible disruption of its pathway may result in disease onset and progression.

## 2. Oleic Acid in Health and Disease

In order to characterize the role of oleic acid in health and disease, numerous studies and clinical trials have been conducted (Table 2). Several studies have highlighted the importance of MUFA in the diet, suggesting that the source and origin of MUFA is fundamental to analyse the beneficial effects of this FA [38]. In Greece, Italy and Spain, olive oil consumption constitutes approximately 60% of MUFA intake, whereas in other countries MUFAs are mainly introduced with meat products [39]. A correlation between MUFA intake and the reduction of cardiovascular heart disease risk have been described [40], whereas a subsequent meta-analysis observed no significant association [41]. These conflicting data could possibly be explained considering the different sources of MUFAs consumed by different cohorts of patients. Llorente-Cortes et al. demonstrated that a MUFA-rich diet reduced the postprandial monocytes inflammatory state linked to metabolic syndrome [42]. Indeed, MUFAs display beneficial effects on insulin sensitivity and type 2 diabetes mellitus [43]. In particular, Vessby et al. observed that the reduction of SFA and the concomitant addition of MUFA to the diet ameliorated insulin sensitivity [44]. The western diet is rich in foods with high SFA abundance such as red meat and processed foods [45]. This fat overload contributes to weight gain and the succeeding inflammation, and SFA induce insulin resistance and type 2 diabetes mellitus [46]. Palmitic acid promotes the synthesis of non-esterified fatty acids (NEFAs), ceramides and reactive oxygen species (ROS), mitochondrial dysfunction and inflammation [47,48]. SFA are able to alter the microbiota composition in the gut as well by up-regulating LPS and toll-like receptor 4 (TLR4) levels [49]. The substitution of palmitic with oleic acid in the diet is able to reverse these detrimental metabolic effects of SFA. Cell culture studies have facilitated the definition of these molecular metabolic changes. In detail, oleic acid enhances mitochondrial oxidation of SFA by increasing triacylglycerol (TAG) and by reducing diacylglycerol (DAG) and ceramide production, thus protecting the cells from inflammation [50].

Furthermore, oleic acid displays the ability to prevent SFA-induced inflammation. In high fat diet fed mice, oleic acid administration ameliorated insulin sensitivity, reduced pro-inflammatory cytokines levels as Interleukin-1β, Interleukin-6 and Tumor Necrosis Factor-α and up-regulated the anti-inflammatory Inteleukin-10 and adiponectin levels [51]. Through this mechanism, oleic acid promotes M2 anti-inflammatory macrophages phenotype, leading to reduction of leukotriene B4 secretion and PTEN expression [52].

The peculiar beneficial role of oleic acid has been observed in several diseases including coronary heart disease, rheumatoid arthritis, and cancer. Oleic acid consumption prevents the risk of developing rheumatoid arthritis by increasing leukotriene A3 levels, a potent inhibitor of pro-inflammatory LTB4 [53]. In colorectal cancer HT-29 cells, oleic acid promotes apoptosis and cell differentiation via the downregulation of cyclooxygenase and Bcl-2 expression [54]. In breast cancer cells, oleic acid reduces the expression of the oncogene Her-2/neu and acts synergistically with anticancer drug trastuzumab [55]. Moreover, in breast tissue cells, oleic acid reduces the entering of lipid peroxidation into the phospholipid membrane of the cells [56].

Interestingly, in animal models, oleic acid is able to induce lung injury miming acute respiratory distress syndrome (ARDS). Oleic acid administration induces direct toxicity to the endothelial cells characterized by endothelial necrosis, epithelial injury and neutrophil infiltration [57,58]. Recently, it has been demonstrated that Liver X Receptor (LXR) activation protects the lungs from oleic acid-induced ARDS by decreasing the inflammatory response and by promoting antioxidant capacity [59]. On the other hand, in lung cancer, oleic acid exerts beneficial effects through promotion of apoptosis, mitosis arrest and cellular differentiation and by inhibiting angiogenesis [60]. In murine models, Piegari et al. have shown that an oleic acid-enriched diet ameliorated animal survival and lung tumour latency, confirming the oleic acid anticancer properties [61].

Finally, oleic acid displays a pivotal role in the development of the brain, the organ with the highest lipid content of the body second to white adipose tissue. In this organ, lipids are essential for the correct homeostasis and alterations in lipid metabolism are linked to neurological diseases [62]. Oleic acid is the only FA synthesized by astrocytes and it acts as a neurotrophic factor for neurons [63]. In astrocytes, albumin uptake and transcytosis via endoplasmic reticulum induces sterol regulatory element-binding protein-1 (SREBP-1) and stearoyl-CoA desaturase expression, causing oleic acid production. The synthesized oleic acid promotes axonal and dendrite growth and induces doublecortin expression, resulting in neuron migration [64,65,66]. These data confirm the pivotal role of oleic acid in astrocyte-neuron crosstalk and further studies should be performed in order to evaluate the role of oleic acid in neurodegenerative disorders. Given the crucial role of oleic acid in brain metabolism, Priore et al. treated C6 glioma cells with oleic acid and hydroxytyrosol and observed a reduction in de novo fatty acid and cholesterol synthesis, a crucial step in human glial cells malignancy [67,68].

## 3. SCD1: The Oleic Acid Producer

SCD is an enzyme anchored to the endoplasmic reticulum, where it catalyzes the biosynthesis of MUFA from SFA, either derived from the diet or synthesized de novo. SCD is Δ9-fatty acyl CoA desaturases that catalyzes the formation of a double bond in the cis-delta-9 position of saturated fatty acyl CoA. The preferred substrates of SCD are palmitic (C16:0) and stearic (C18:0) acids, which are converted into palmitoleic (C16:1 n-7) and oleic (C18:1 n-9) acids, respectively [69] (Figure 1). These products are the most abundant MUFA incorporated into different lipid species, including triglycerides, wax esters, cholesterol esters, and phospholipids [70]. The ratio of SFA to MUFA is important to modulate phospholipid composition, and an unbalanced ratio toward SFA production is linked to multiple pathological conditions including obesity, diabetes, cardiovascular and neurological diseases, and cancer [71]. Moreover, MUFAs serve as mediators of signal transduction and cellular differentiation [71]. Thus, the expression of SCD is highly important in both physiological and pathological conditions and SCD is consequently tightly controlled at both transcriptional and post-translational level. Numerous dietary and hormonal signals and several transcription factors such as LXR, SREBP1C, carbohydrate response element binding protein (ChREBP), peroxisome proliferator activated receptor (PPAR), and estrogen receptor (ER) are involved in SCD transcriptional control [72,73].

SCD is a highly conserved enzyme with multiple isoforms sharing similarities in protein sequences [74]. To date, two human isoforms (1 and 5) and four mouse SCD isoforms (SCD1-4) have been identified [75,76]. Despite a distinct distribution pattern, the different isoforms share the equal enzymatic function. Particularly, in both human and mice, SCD1 represents the predominant isoform and it is ubiquitously expressed among tissues, with constitutively high levels in lipogenic tissues, such as liver and white adipose tissue [74]. Contrarily, SCD5 is unique to primates and is highly expressed in brain and pancreas [76]. In adult mice, Scd2 isoform is ubiquitously expressed in most tissues except for the liver, whereas Scd3 and Scd4 expression is more restricted [77,78,79].

The existence of multiple SCD1 isoforms, sharing a high sequence homology and catalyzing the same biochemical reaction, poses difficulties in order to distinguish the role of each single isoform and its metabolic contribution. To this end, substantial insights into SCD1 functions have been gained by the use of specific mouse models, in which the expression of SCD1 is downregulated or its activity is inhibited. Several genetically engineered whole body and tissue specific SCD1 knockout models have been utilized [80] as well as Asebia mice, characterized by a whole-body deficiency of SCD1 due to a spontaneous mutation within the SCD1 gene [81]. Additionally, mice treated with antisense oligonucleotides (ASO) against SCD1—which reduces SCD1 expression in liver and adipose tissue—have been extensively employed [82]. Finally, various selective inhibitors targeting SCD1 activity have enabled the clarification of the role of this enzyme. Overall, these models demonstrate that SCD1 is a master regulator of lipid metabolism and is deeply involved in body weight regulation [83,84]. This is specifically highlighted by the resistance to diet-induced obesity and hepatic steatosis observed in SCD1 deficient mice [85]. These effects are predominantly derived from an increase in FA oxidation and thermogenesis, as well as a reduced lipid synthesis. The increased FA oxidation is mainly explained by a defect of skin permeability in SCD1KO mice which is causing heat leak and water link. The increased lipid oxidation is mediated by the induction of the AMP-activated protein kinase (AMPK) [86], resulting in phosphorylation and inactivation of ACC. The inactivation of this enzyme reduces the cellular levels of malonyl-CoA, a substrate for FA biosynthesis, which in turn suppresses FA oxidation by inhibiting the mitochondrial carnitine palmitoyltransferase 1 (CPT1) shuttle system, which controls the import and oxidation of FA in mitochondria. Malonyl-CoA reduction in SCD1KO mice relieves the inhibition of CPT1 by directing FA into mitochondria where they are subsequently oxidized. In SCD1KO mice, the up-regulation of whole-body thermogenesis in brown adipose tissue (BAT) is mediated by the activation of PGC-1α and uncoupling protein-1, which uncouples oxidative respiration from ATP synthesis, resulting in dissipation of energy as heat [87]. These modulations result in an increased rate of basal thermogenesis and, consequently, of whole-body energy expenditure.

Although many efforts have been made utilizing total body SCD1KO mice, the confirmation that these mice display altered skin permeability enhancing energy expenditure and protection from high fat diet-induced obesity [88], has led to the generation of tissue specific SCD1KO mice in order to study the contribution of this enzyme to metabolic disorders and related diseases.

### SCD1 in Pathological Conditions

The central contribution of FA to cellular inflammation has been extensively demonstrated [89]. Recently, SFA have been recognized as a potent proinflammatory factor in multiple cell types, including macrophages, myocytes, endothelial cells, adipocytes, and β-cells [73]. Indeed, SFA may directly stimulate the inflammatory process or can be metabolized into lipid intermediates, as ceramides and diacylglycerols; two powerful proinflammatory factors. Correspondingly, the complete loss of SCD1 expression has been implicated in liver dysfunction and several inflammatory diseases such as dermatitis, intestinal colitis and atherosclerosis [73,90,91,92].

Inflammation caused by abnormal levels of lipids remaining unresolved for an extended period of time may lead to cellular stress and dysfunction, resulting in lipotoxicity, one of the major causes of pathologies including obesity, insulin resistance and cardiovascular disease [93,94]. The lipogenic enzyme SCD1 exerts a crucial role in the development of obesity and related conditions, such as insulin resistance [95]. The involvement of SCD1 in the regulation of obesity has been demonstrated by the increase of its activity in the adipose tissue of various animal models of obesity, such as leptin-deficient ob/ob mice, which develop an obese phenotype early on in life [96]. Leptin is an adipocyte-derived liporegulatory hormone controlling lipid homeostasis in non-adipose tissues, at least partly, by targeting hepatic SCD1 activity [96]. Indeed, double mutant abJ/abJ; ob/ob mice, obtained by crossing heterozygous Asebia (abJ/+) mice with heterozygous ob/+ mice, exhibited a dramatic reduction in body weight compared to ob/ob littermates despite a high consumption of food. The deletion of SCD1 reverted the hypometabolic phenotype, the hepatomegaly and the hepatic steatosis typical of the ob/ob mice as well [96]. Overall, these evidences demonstrate that SCD1 is required for the development of the obese phenotype of ob/ob mice and that downregulation of SCD1 is a fundamental part of leptin’s metabolic effects.

In addition, SCD1 global depletion increases insulin sensitivity and glucose utilization [97]. This might be ascribable to a reduced Akt activity and insulin receptor substrate phosphorylation (mediated by the activation of protein kinase C as well) with an impaired GLUT4 translocation to the plasma membrane [98,99]. As SCD1 inhibition may induce insulin sensitivity in muscle [100] and BAT [101], it is therefore plausible to explore the use of SCD as a potential therapeutic target for the treatment of insulin resistance and diabetes.

Overall, these metabolic diseases are characterized by a metabolic shift toward biosynthetic reactions aimed to supply new lipid macromolecules. Interestingly, enhanced FA biogenesis is a hallmark of cancer as well. Whereas in obesity and diabetes, the elevated lipid biosynthetic activity result in increased production lipids for energy storage, most of the newly synthesized lipid products are utilized for membrane biogenesis in cancer [102]. The metabolism of cancer cells is characterized not only by an increased biosynthesis of macromolecules including lipids with a concomitant suppression of their catabolic pathways [103], but by an abnormally high rate of aerobic glycolysis that provides metabolites as well, such as citrate and glycerol, used for the de novo synthesis of cellular lipids [104]. The parallel activation of these two processes in cancer is coupled to the conversion of SFA into MUFA. Although SFA are the main products of glucose-derived FA synthesis, an increased content of MUFA is found in transformed and cancer cells and tissues as well [105].

Recent analyses have revealed a fundamental contribution of SCD1 to the metabolic and signaling pathways closely related to cell replication, survival and tumorigenesis [106]. The molecular regulation of lipogenesis by SCD1 in cancer cells involves different mechanisms (Figure 2). Beside the well-known role in modulating the de novo lipogenesis network, SCD1 can provide MUFAs for lipid macromolecule formation as well as regulate signaling pathways controlling the expression and the activity of the main enzymes involved in lipid biosynthesis [105]. This is exemplified by the well characterized control of ACC function mediated by SCD1. Since ACC activity is repressed by SFA, the involvement of SCD1 in removal of SFA through transformation into MUFA mediates the allosteric stimulation of ACC activity. SCD1 inactivates AMPK phosphorylation with a consequent inhibition of its target activity as well, such as ACC, FASN and hydroxymethylglutaryl-coenzyme A reductase (a critical enzyme in cholesterol synthesis) [107]. Lastly, SCD1 activation has been shown to promote the Akt pathway, with potential positive effects on the transcription of lipogenic enzymes. This is highly interesting, as Akt can exert a direct action on cellular proliferation by stimulating a group of transcriptional factors, such as the SREBP family, thus inducing lipid synthesis and membranes formation [108]. It is however important to note that Akt via glycogen synthase kinase 3-beta may play an indirect action on cell growth as well by downregulating crucial regulators of cell cycle progression, such as cyclin D1 and its transcriptional activator, β-catenin [109].

Due to the crucial role of lipid biosynthesis in tumor cells, SCD1 deletion could modulate several phenotypic features of cancer. Specifically, SCD1 enhances cancer cell mitogenesis and survival, increases tumorigenic capacity and tumor cell invasiveness [110,111]. Even if the antiproliferative effect of SCD1 inhibition is common to many neoplastic cell types including melanoma and lung, bladder, and breast cancer [112,113,114,115], the role of SCD1 in tumorigenesis cannot be generalized. This is demonstrated by the consistently reduced SCD1 activity prostate carcinomas [116]. Although the use of SCD1 as a target for novel pharmacological approaches in cancer interventions deserve interest, it has to be referred to a specific tumor type.

## 4. SCD1 in the Gut-Liver Axis

The liver plays a central role as a metabolic hub, where multiple biochemical processes converge to render food nutrients available to the rest of the body. However, an unbalanced diet—rich in lipid and carbohydrates—together with a sedentary lifestyle, may severely compromise the hepatic health status. Indeed, it has been shown that a high energy intake with a concomitant low energy expenditure resulted in hepatic lipid accumulation, one of the key features of non-alcoholic fatty liver disease (NAFLD). NAFLD is the most common disease of the Western Countries, characterized by a spectrum of conditions that vary from hepatic steatosis to severe form of non-alcoholic steatohepatitis (NASH), which may eventually progress toward cirrhosis and end-stage liver diseases [117]. Despite the large amount of studies focused on the impact of hepatic metabolic alterations leading to the toxic effects of excess lipids and promotes liver diseases, it is now clear that metabolites from other organs may account for disease progression, such as adipose tissue and gut, as well. Whereas the communication between the adipose tissue and the liver has been extensively investigated, the gut-liver cross-talk exploration is just at the beginning. Not only the liver can influence the gut phenotype via bile acids or metabolites, but the gut harboring microbiota and secretory factors as well can affect the hepatic health status and promote lipid accumulation. However, accumulating data show that the total amount of lipids is not the major determinant of lipotoxicity, but that specific classes of lipids (palmitic acid, cholesterol, ceramides) promote cellular damage and disease progression [118]. In this context, by exploiting the role of SCD1 and its product oleic acid in the modulation of hepatic and intestinal metabolism, we discuss our understanding of the biological impact of gene-nutrients interaction, providing a key insight into the liver and gut diseases.

### 4.1. SCD1 and the Liver

Lipids are essential for several processes that involve cellular and tissue homeostasis. Expression of lipogenic genes, including SCD1, is essential for hepatic maturation and function [119]. Indeed, human embryonic stem cells (hESC) primarily depend on SCD1 activity for their survival and metabolic processes. SCD1 inhibition in hESC resulted in an altered SFA to MUFA ratio, characterized by oleate depletion leading to endoplasmic reticulum stress, unfolded protein response and translational attenuation, which collectively promotes apoptosis [120]. Moreover, human induced pluripotent stem cells (hiPSC) treated with SCD1 inhibitor displayed a suppression of mature hepatic marker products and TG accumulation that were promptly reversed by oleate supplementation [121]. Collectively, these data indicate that SCD1 and its product are necessary for differentiation of stem cells into mature hepatocytes. Noteworthy, SCD1 expression appeared to be dispensable in mice embryos liver, while the Scd2 isoform (SCD5 in human) is crucial at this stage. Scd2 hepatic expression decreased in the liver before weaning, and it is replaced by SCD1, which is highly activated in adult liver [77].

Genetically engineered whole body and liver specific SCD1 knockout animal models as well as mice—in which SCD1 activity is suppressed—has provided substantial insights into hepatic SCD1 functions [81,82,85,122,123,124]. Altogether, this research described the central role of SCD1 in metabolic disorders characterized by altered lipid metabolism (Figure 3).

Whole body SCD1 ablation resulted in global modifications of gene expression and metabolic processes that cause loss of body fat and increased insulin sensitivity. Specifically, the transcription of several genes involved in de novo lipogenesis were downregulated in the liver of SCD1KO mice, accompanied by an upregulation of genes involved in lipid β-oxidation [122]. When challenged with dietary (high-fat and high-carbohydrate diet) or genetic (leptin deficient and agouti induced) models of obesity, mice deficient for SCD1 displayed protection from fat accumulation and hepatic steatosis, thus highlighting the requirement of SCD1 to fully develop the obese phenotype peculiar of these models [82,96,122,125,126]. Indeed, SCD1 expression and activity is highly elevated in the liver of obese animal models, as the ob/ob mice and the Zucker rats [96,127]. SCD1KO mice fed with very low fat-diet for 10 days displayed decreased triglycerides content together with severe hepatic damage, which is promptly rescued by oleic acid supplementation, ultimately reversing the hepatotoxic effect. The insufficient MUFA levels due to the unsaturated fat deficient diet and the ablation of SCD1 results in dramatic changes of hepatic gene expression, identified by the downregulation of lipid metabolism and detoxification programs together with the concomitant upregulation pathways involved in ER stress and inflammation [128,129]. The maintenance of adequate MUFA levels through SCD1 activity or oleate ingestion is therefore fundamental for preservation of liver health status.

The observations obtained from global SCD1 ablation led to the generation of liver specific SCD1KO (LSCD1KO) mice using Cre-Lox system, in order to investigate whether the hepatic SCD1 might be responsible for the observed phenotypes. Differently from whole body SCD1KO, mice lacking SCD1 in the liver were not protected from both genetic and diet-induce obesity, although they displayed lowered content of hepatic MUFA and triglycerides [130]. The loss of SCD1 alone in the liver is consequently not sufficient to elicit the hypermetabolism and energy expenditure needed to compensate for the energy intake of high fat diet. This is most likely due to the skin defective composition of SCD1KO mice. Indeed, skin-specific deletion of SCD1 display augmented energy expenditure and are protected from high fat diet-induced obesity, thereby recapping the hypermetabolic phenotype of global Sc11 deficiency [88]. Conversely, LSCD1KO mice are protected against high carbohydrates induced liver steatosis; the consumption of high sucrose-very low-fat diet in LSCD1KO mice resulted in hypoglycemic phenotype and low induction of lipogenic programs [85]. It can be speculated whether the low amount of MUFA in the liver of LSCD1KO mice induces the utilization of carbohydrates for heat generation, and not for ATP synthesis, thus explaining the hypoglycemia observed. Dietary Oleate supplementation normalized the triglycerides secretion and the hepatic gluconeogenesis and lipogenesis pathways, suggesting that hepatic SCD1 activity and adequate oleate content are required for carbohydrates induced fat accumulation [85]. Collectively, these data indicate that SCD1 inhibition in extrahepatic tissues is necessary to confer resistance to high fat diet-induced weight gain and hepatic steatosis [88]. MUFA derived from endogenous synthesis in other tissues are likely compensating for the reduced liver MUFA production in LSCD1KO mice.

Although loss of hepatic SCD1 expression confers beneficial metabolic effect in terms of protection from fat accumulation, several studies have pointed out that SCD1 and its product oleate are necessary to preserve liver function in a multiplicity of stressful circumstances. Mice fed with a Methionine and Choline Deficient (MCD) diet—a model of steatohepatitis—displayed diminished expression of SCD1 in the liver [131,132]. When challenged with an MCD diet, SCD1KO mice showed lower steatosis, but increased induction of hepatocytes apoptosis as well as liver injury and fibrosis. Supplementation with MUFA prevented the MCD-induced injuries, which is an indication that the steatohepatitis phenotype observed is mostly due to the accumulation of SFA promoting apoptosis [132]. Moreover, hamster treated with sterculic oil—a potent inhibitor of SCD1 activity—and fed with cholesterol enriched diet displayed liver damage together with ALT increase and hypercholesterolemia [124]. By reducing the SCD1 activity, the low level of its product oleic acid failed to be funneled into hepatic cholesteryl ester, eventually not conferring hepatic protection from toxic cholesterol and derivatives. SCD1 is therefore able to prevent liver injury, mostly by partitioning the excess of lipids into MUFAs that can be safely stored in the liver. However, in a mouse model of concavalin A-induced hepatitis, the expression levels of SCD1 were upregulated together with increased lipid accumulation in the hepatocytes. Correspondingly, SCD1KO mice treated with concanavalin A are protected to steatohepatitis in a leptin dependent manner [123]. The conflicting results on SCD1 role in the liver may be related to the different approaches utilized to induce hepatic damage: whereas dietary models mainly affect hepatocytes, concanavalin A-induced injury is primarily driven by the activation and recruitment of NK-T cells in the liver [133]. Other studies are consequently needed to fully dissect the SCD1 contribution to liver function.

Although the contribution of SCD1 to NAFLD is still not fully depicted, recent studies have highlighted the important association between SCD1 expression and hepatocarcinoma (HCC) progression. The expression of SCD1, as well as genes involved in FA metabolism, is upregulated in HCC, and is strictly related to a poor prognosis [134,135,136,137]. Moreover, MUFA produced by SCD1 enzymatic activity amplifies Wnt-βCatenin signaling in HCC cells, thus leading to tumor growth [135]. SCD1 inhibition reduced cell viability, induced apoptosis and autophagy and sensitized cells to sorafenib, a standard treatment for HCC patients in advanced stages [134,136,138]. Most of these studies have been conducted on human samples, cell cultures and xenograft, and the in vivo evidence able to display the huge complexity of organ-to-organ communication is yet lacking.

Overall, these studies have suggested that SCD1 expression and the activity or oleic acid supplemented with the diet or originated from other tissues subsequently delivered to the liver are fundamental for the proper development and function of the liver. Although the analysis regarding SCD1 in HCC seems to offer a different scenario, where SCD1 activity contributes to tumor progression, they depict an established situation without taking into account the contribution of SCD1 to tumor onset and the possible beneficial effect of oleic acid supplementation. Further studies focused on the part played by SCD1 in the first steps of liver tumor formation as well as on the role in HCC development of oleic acid synthetized by other tissues are required.

### 4.2. SCD1 and the Gut

Different from the liver, the role of intestinal SCD1 has been less characterized. The few studies conducted depict an incoherent scenario, where SCD1 may both confer protection or worsen intestinal diseases (Figure 4). Here, we will recapitulate the principal investigations—even if further in vivo studies are definitely required.

The analysis of cancerous specimens collected from patients with stage II colon cancer (CRC) revealed that an elevated expression of SCD1 together with other three genes involved in lipid metabolism (ABCA1, ACSL1, AGPAT1) delineate a high-risk group of patients, characterized by a worse clinical outcome [139]. Indeed, the suppression of SCD1 activity in colon cancer cell lines via specific inhibitor or siRNA had cytotoxic effects, which resulted in the interruption of tumor growth [106,140]. Specifically, in vitro studies in CRC cell lines observed that SCD1 repression by A939 inhibitor delayed tumor growth and promoted apoptosis mainly through mitochondrial disfunction and ROS accumulation. Intriguingly, supplementation with oleic acid reversed the inhibition of cellular proliferation [141]. Although it would be interesting to examine if SCD1 activity and oleic acid supplementation lead to the same results using non-tumoral cell lines as well, these data overall point to a tumor promoting role of SCD1 and its product—oleic acid.

However, the comparison between adjacent normal tissue and cancerous tissue of a cohort of CRC patients revealed that SCD1 expression and activity are decreased in cancerous specimens. Indeed, cancerous tissue displayed higher content of SFA and a lower amount of MUFA compared to normal one [142]. Moreover, studies performed on SCD1 null mice indicated that they were more susceptible to dextran sodium sulfate (DSS)-induced colitis mouse model, largely used to mimic human inflammatory bowel diseases [92]—even if a later study has attributed this phenotype to the greater DSS-total intake of SCD1KO mice compare to control [143]. However, the supplementation of the diet with oleic acid rescued the DSS-phenotype, thus indicating that SCD1 activity and its product oleic acid are necessary to limit the proinflammatory responses to exogenous challenges in the mice gut [92]. Accordingly, mice with specific ablation of SCD1 (iSCD1^-/-^) in the intestinal epithelium presented an increased expression of inflammatory markers and crypt proliferative genes compared to the control group [144]. These results appear in net contrast with those describing less dyslipidemia and systemic inflammation in iSCD1^-/-^ mice crossed with LDLR null mice [145]. However, the use of a LDLR null background makes any comparison between the two studies hazardous. Interestingly, when iSCD1^-/-^ mice are backcrossed with Apc^Min/+^ mice, a genetic model of CRC highly susceptible to spontaneous intestinal adenoma formation, they developed and larger and more tumors compared to the mice expressing SCD1 in the gut. However, the consumption of an oleic acid-enriched diet ameliorated the phenotype, reducing intestinal inflammation and tumor development [144]. Overall, these data indicate that the SCD1 activity in the gut is fundamental to maintain the intestinal epithelial homeostasis and to protect against CRC.

### 4.3. SCD1 in the Gut-Liver Cross Talk

The close relationship between the liver and the gut under the light of SCD1 have been fairly characterized. Gut microbiota may variously affect the function of SCD1 in mice liver. Indeed, hepatic SCD1 levels are substantially reduced in germ free and microbiota ablated mice, thus indicating that SCD1 is partly regulated via gut microbial metabolites [146]. Moreover, mice deficient of TLR5—a flagellin receptor required for gut microbiota homeostasis—display a three-fold higher bacterial load and a concomitant increased expression of hepatic SCD1. When microbial homeostasis is altered, the high amount of single chain FA products generated by microbiota contribute to hepatic lipogenesis, rendering TLR5-deficient mice more prone to develop metabolic syndrome [146]. Multi-omics analysis of mice liver and gut microbiota revealed that the level of hepatic FA desaturation by SCD1 is strictly dependent on the microbial load [147]. In a similar way, the liver can affect the gut homeostasis as well. In this organ, SCD1 is necessary to synthetize oleoyl lysophosphatidylcholine, which protects from the gut inflammation processes [92]. Although more studies are needed to better clarify the role of SCD1 in the close connection between the liver and the gut, these results suggest the importance of considering the organism as a whole, particularly when we discuss about diseases strongly associated with lipid metabolism.

## 5. Conclusions

Intensive studies have demonstrated the fundamental contribution of oleic acid to health status maintenance. At the same time, the rate limiting enzyme for its production, SCD1, has emerged as a key regulator in serious diseases associated with inflammation and stress. Despite the large number of data, the role of SCD1 in these disorders is complex and, in many cases, conflicting results have been obtained, most likely due to variation of the experimental settings as well as of the strategies for the suppression of SCD1 activity (Table 3). Therefore, the mechanism of SCD1 in the regulation of liver and intestinal diseases needs additional investigations. Further studies on the role of SCD1 and its product oleic acid in metabolic homeostasis are needed in order to enhance our knowledge on the function of this enzyme. Since it is clear the activity of SCD1 in one organ may influence other districts—as in the case of gut-liver axis—the future investigations should point to the consideration of the whole organism to better clarify the importance of SCD1 in lipid metabolism and inflammatory pathways strictly connected with the disease progression.

## Figures and Tables

**Figure 1 nutrients-11-02283-f001:**
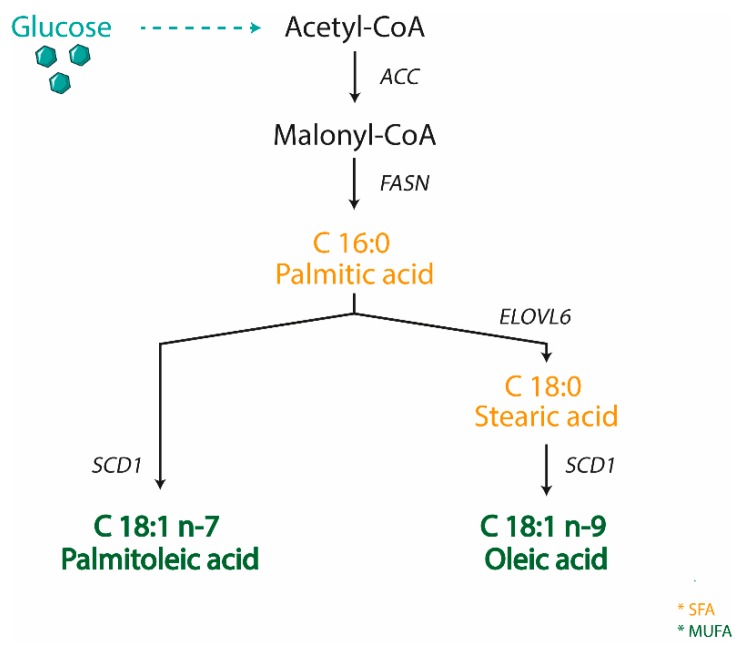
Scheme representing the biosynthesis of MUFA in animals. Stearoyl CoA Desaturase 1 (SCD1) catalyzes the rate-limiting step for the conversion of saturated fatty acid (SFA) into monounsaturated ones (MUFA). (ACC—acetyl CoA carboxylase; FASN—fatty acid synthase; ELOVL6—fatty acid elongases 6).

**Figure 2 nutrients-11-02283-f002:**
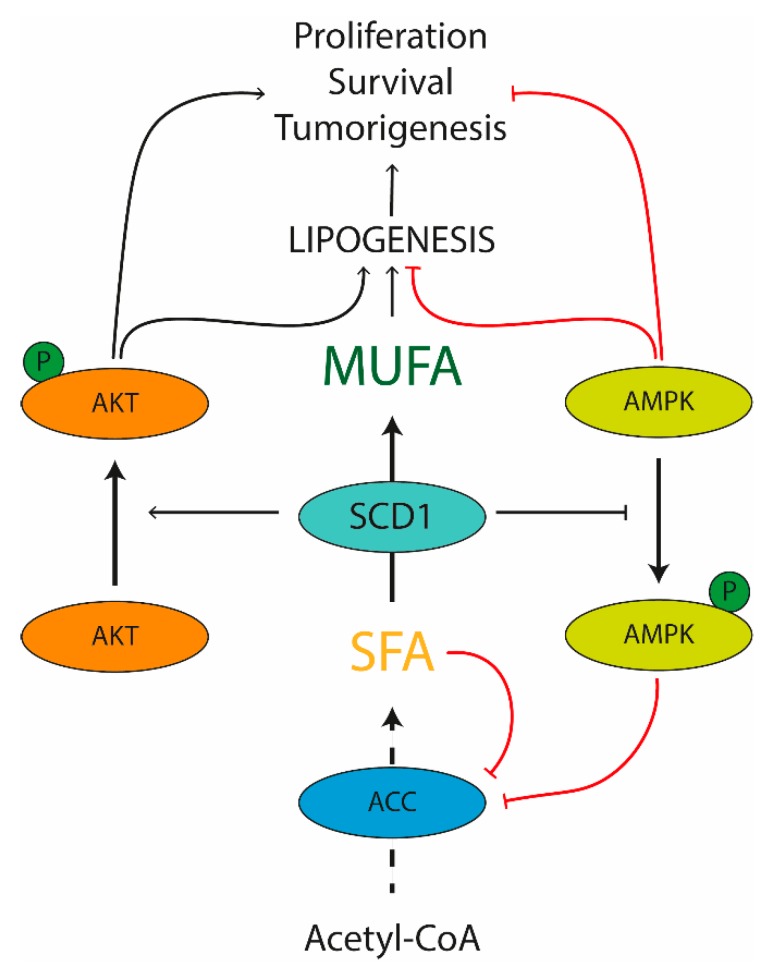
A theoretical model of metabolic control of lipogenesis by SCD1 in cancer cells. SCD1—stearoyl-CoA desaturase 1; ACC—acetyl-CoA carboxylase; AMPK—AMP-activated kinase; MUFA—monounsaturated fatty acids; SFA—saturated fatty acids.

**Figure 3 nutrients-11-02283-f003:**
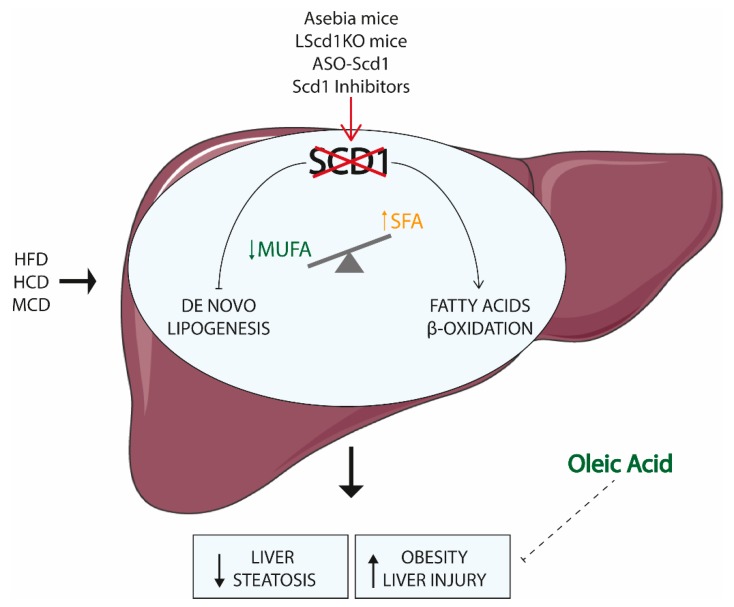
SCD1 inhibition in non-alcholic fatty liver disease (NAFLD). Different methodologies have been used to suppress the stearoyl CoA desaturase 1 (SCD1) activity in the liver, ranging from tissue specific mouse model (LSCD1KO) to Asebia mice, homozygous for naturally occurred mutation which result in the lack of SCD1, to specific inhibitor or antisense oligonucleotide (ASO). At the same time, different disease models have been employed to dissect the role of SCD1 in hepatic disorders. The main one consisted in diet administration, such as a high fat diet (HFD), high carbohydrates diet (HCD) and methionine-choline deficient diet (MCD). The hepatic suppression of SCD1 expression resulted in an altered saturated (SFA) to monounsaturated fatty acid (MUFA) ratio, with the concomitant decreased de novo lipogenesis programs and increased fatty acids β-oxidation pathways. Although these animals are protected from liver steatosis, they normally display liver injury, which is promptly rescued by oleic acid endogenous (from other tissues, such as intestine or white adipose tissue) or exogenous (from diet) supplementation.

**Figure 4 nutrients-11-02283-f004:**
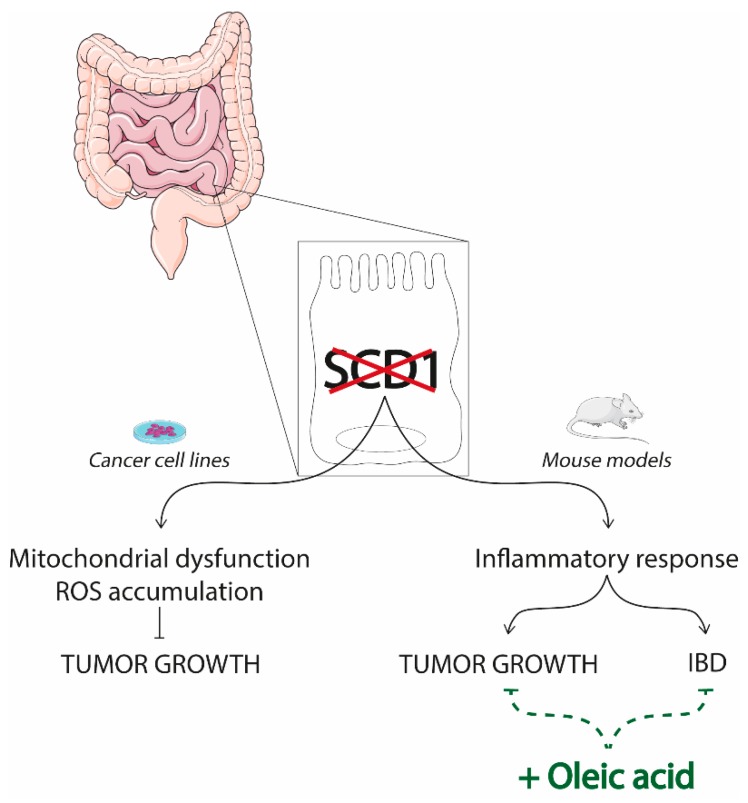
SCD1 inhibition in gut disorders. stearoyl CoA desaturase 1 (SCD1) suppression in intestinal cells offers a divergent scenario. Studies conducted in cancer cell lines indicated that SCD1 suppression inhibits tumor growth by increasing mitochondrial dysfunction and reactive oxygen species (ROS) accumulation. On the other side, investigations carried out using mouse models, demonstrated that the loss of SCD1 promotes an inflammatory state that worsens inflammatory bowel disease (IBD) as well as cancer growth and progression. Intriguingly, the consumption of oleic acid enriched diet reverses the disease phenotype.

**Table 1 nutrients-11-02283-t001:** Stearoyl-CoA desaturase 1 (SCD1) clinical trials.

Trial Identifier	Trial Phase (Status)	Title	Intervention
NCT02647970	Completed	Stearoyl-CoA Desaturase and Energy Metabolism in Humans	Behavioral: PUFA‒Cys/Met diet Behavioral: SFA+Cys/Met diet
NCT03572205	Completed	Fatty Acid Desaturase Gene Locus Interactions with Diet (FADSDIET2)	Dietary Supplement: LA Dietary Supplement: ALA
NCT03282253	Not yet recruiting	Elevated Stearoyl-CoA Desaturase-1 Expression Predicts the Disease Severity of Severe Acute Pancreatitis	
NCT02543216	Completed	Gene–Diet Interactions in Fatty Acid Desaturase 1 Gene	Dietary Supplement: Sunflower oil
NCT03842891	Completed	Genetic Variants Modulate Association Between Dietary n-3 LCPUFAs and DHA Proportion in Breast Milk	
NCT01661764	Completed, Has Results	Fish Oil Supplementation, Nutrigenomics and Colorectal Cancer Prevention	Drug: Eicosapentanoic acid and docosahexanoic acid Drug: Oleic Acid
NCT02337231	Completed	Botanical Oils Study to Determine Genetic Differences in the Way Your Body Processes Fats in Edible Oils	Dietary Supplement: soybean oil and borage oil

**Table 2 nutrients-11-02283-t002:** Oleic acid clinical trials.

Trial Identifier	Trial Phase (Status)	Title	Intervention
NCT00715312	Completed	Effect of Oleic Acid on Inflammation Markers and Blood Lipid Metabolites: A Randomised, Double-Blind, Crossover Study	Novel Olein
NCT01042340	Completed	Energy Dense Oleic Acid Formula to Geriatric Patients	Calogen®–an energy dense oleic acid-based formula
NCT01124487	Completed	The Acute Effects of Oleic Acid Enriched diets on Lipids, Insulin Sensitivity and Serum Inflammatory Markers	Dietary Supplement: The acute effects of dietary fat on lipid profile, insulin sensitivity and inflammatory markers
NCT02029833	Completed	Canola Oil Multi-Centre Intervention Trial II	Other: Regular Canola Oil Other: High Oleic Canola Oil Other: Western Type Diet–Common Dietary Oils
NCT03054779	Completed	Canola Oil Multi-Centre Intervention Trial II	Other: Canola Oil Other: High oleic acid canola oil Other: Western diet oil combination
NCT02993380	Completed	Effect of Olive Oil on Erythrocyte Membrane Fatty Acid Contents in Hemodialysis Patients	Dietary Supplement: Stir-fried olive oil group Dietary Supplement: Natural olive oil group
NCT00529828	Completed	Health Effects of CLA Versus Industrial Trans Fatty Acids	Procedure: Consumption of CLA enriched food
NCT01259999	Completed	Energy Dense Formula to People Living in Old Peoples Home	Dietary Supplement: Calogen extra strawberry
NCT00059254	Completed	Differential Metabolism of Dietary Fatty Acids	Dietary Supplement: Oleic acid (OA) Dietary Supplement: Palmitic Acid (PA)
NCT01996566	Completed	Fatty Acid Taste thresholds: Caproic, Lauric, Oleic, Linoleic, Linolenic	

**Table 3 nutrients-11-02283-t003:** Principal studies assessing the role of SCD1 in liver and intestine.

Type of Study	SCD1 Status	Phenotype	Ref
*In vivo*			
SCD1KO mice	Whole body SCD1 deletion	Protection from HFD and HCD-induced adiposity and hepatic steatosis. High susceptibility to DSS-induced gut inflammation	[92,122]
Asebia (abJ/abJ) mice	Lack of SCD1 for a naturally occurred mutation	Protection from liver steatosis and adiposity	[82,96]
LSCD1KO mice	Liver specific SCD1 deletion	Protection from HCD-induced adiposity and hepatic steatosis. Susceptibility to HFD-induced obesity and hepatic steatosis.	[85]
LASCD1KO mice	Adipose and liver combined SCD1 deletion	Susceptibility from diet induced obesity	[130]
iSCD1KO mice	Intestinal specific SCD1 deletion	Susceptibility to CRC when crossed with ApcMin mice and fed with oleic acid deficient diet	[144]
Hamster treated with SCD1 inhibitor	Inhibition of SCD1 activity	Liver protection from cholesterol enriched diet and susceptibility to atherogenic risk	[124]
*Ex Vivo*			
HCC specimens	High SCD1 expression	Shorter disease-free survival and sorafenib resistance in HCC	[136]
CRC specimens	Low SCD1 activity		[142]
CRC specimens	High SCD1 expression	Worse clinical CRC outcome	[139,141]
*In vitro*			
Cell culture treated with SCD1 inhibitor	Inhibition of SCD1 activity	Suppression of tumor cell proliferation and apoptosis induction	[106,141]

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
