# Peer review of "Role of Oleic Acid in the Gut-Liver Axis: From Diet to the Regulation of Its Synthesis via Stearoyl-CoA Desaturase 1 (SCD1)"

_nutrients, 2019, doi:10.3390/nu11102283_

Round 1

Reviewer 1 Report

Comments and suggestions to the authors
ï‚· An abbreviation list could be useful for a better reading of the manuscript.
ï‚· L59: reference is misplaced, the cited link must be included in the reference list, with its proper information, and correctly cited in text.
ï‚· In the introduction the role of edible oils in health is stated. The authors mentioned EVOO, SFO, palm oil and fish oil. Nonetheless, at this moment there are several other edible oils that play an important role in health, especially the ones that on its composition have important amounts of MUFA and PUFA (mainly omega-3/6), or even oleic acid. This part of the manuscript should include at least a paragraph that mentions this information, based on the recent scientific evidence.
ï‚· L104: same as in L59. Please correct.
ï‚· Throughout the manuscript if abbreviations are used the authors should be careful to apply them in all the text, i.e. L156 <…saturated to monounsaturated fatty acids…> it could be used SFA and MUFA.
ï‚· All the bibliographic research is accurate and well explained. However, for a review manuscript, I consider that there are several lacks in the order and method to display the information. The importance of oleic acid must be summarized and categorized, preferably in tables, with the recent and relevant studies performed in vitro, cell cultures, animals, and humans’ models. Explaining on each section the food science, nutrition, and/or pharmacology-biochemistry-physiology evidence. And in the same way for Scd1 studies.
ï‚· Figures are way too abstracted. In text there are many important pathway explanations that are not later represented in figures. Authors should include figures that clearly state the information that is previously explained in text.
ï‚· Reference list presents a few mistakes in homogeneity. All references should be carefully revised in journal names, etc.

Author Response

POINT-BY-POINT REPLY TO REVIEWERS’ COMMENTS

nutrients-596305

“Role of oleic acid in the gut-liver axis: from diet to the regulation of its synthesis via Stearoyl-CoA desaturase 1 (Scd1)”

Nutrients

Reviewer 1

We would like to thank Reviewer 1 for the effort and time he/she put in revised our manuscript. We have addressed all of the comments offered by Reviewer 1 and we honestly feel that his/her insightful suggestions helped us in improving the quality of the manuscript. Below, we outline how we have handled each of Reviewer 1’s comments. We reiterate each suggestion in italics.

Comments and suggestions to the authors

An abbreviation list could be useful for a better reading of the manuscript.

We really appreciate this suggestion. We are now providing an abbreviation list with all the most recurrent acronyms.

L59: reference is misplaced, the cited link must be included in the reference list, with its proper information, and correctly cited in text.

The link has been now included in the reference list.

In the introduction the role of edible oils in health is stated. The authors mentioned EVOO, SFO, palm oil and fish oil. Nonetheless, at this moment there are several other edible oils that play an important role in health, especially the ones that on its composition have important amounts of MUFA and PUFA (mainly omega-3/6), or even oleic acid. This part of the manuscript should include at least a paragraph that mentions this information, based on the recent scientific evidence.

We are grateful to Reviewer 1 for raising this very important point. In the new version of the manuscript, we added a paragraph mentioning edible oils with crucial role in health, a part from the previous mentioned.

L104: same as in L59. Please correct.

The link has been now included in the reference list.

Throughout the manuscript if abbreviations are used the authors should be careful to apply them in all the text, i.e. L156 <…saturated to monounsaturated fatty acids…> it could be used SFA and MUFA.

We apologize for the inconvenience. We corrected the text accordingly.

All the bibliographic research is accurate and well explained. However, for a review manuscript, I consider that there are several lacks in the order and method to display the information. The importance of oleic acid must be summarized and categorized, preferably in tables, with the recent and relevant studies performed in vitro, cell cultures, animals, and humans’ models. Explaining on each section the food science, nutrition, and/or pharmacology-biochemistry-physiology evidence. And in the same way for Scd1 studies.

We are extremely grateful for the Reviewer suggestion, that led us to include three tables in the new version of the manuscript. We believe that this additional information helped us to improve the overall quality of the manuscript. As suggested, in order to highlight the importance of oleic acid for human health, in Table 2 we reported all the clinical trials regarding the oleic acid, whereas Table 1 and Table 3 are dedicated to illustrate the importance of Scd1 in health and disease. Particularly, Table 1 reported all the clinical trial about the master regulator of oleic acid synthesis, SCD1, and in Table 2 we indicated all the recent and relevant studies performed in vivo, ex vivo and in vitro on SCD1.

Figures are way too abstracted. In text there are many important pathway explanations that are not later represented in figures. Authors should include figures that clearly state the information that is previously explained in text.

We apologize for the lack of clarity in the figures. Since many of the results regarding the contribution of Scd1 to liver and intestinal diseases do not offer a unique scenario, it is for us really difficult to include all the information provided in the text in one figure, since it may result in a really complicate and intricate picture. However, we agree with the Reviewer that a figure recapitulating the principal important pathways regulated by SCD1 is of fundamental importance. Therefore, we are now providing a new figure including all the main molecular process regulated by SCD1 in cancer cells. We thank the Reviewer for his/her constructive comments.

Reference list presents a few mistakes in homogeneity. All references should be carefully revised in journal names, etc.

We apologize for this inconvenience. We modified the text accordingly to the Reviewer’s indications.

Reviewer 2 Report

Introduction

line 42 - the correct reference for 18;3n-3 is α-linolenic acid, not linolenic acid

line 44- need to be more specific here, it is the inclusion of the 2 essential fatty acids that are important - linoleic acid and α-linolenic acid. No other fats have been shown to have essentiality - this is different from fats being linked to incidence of disease as you are describing here. Please rectify.

lines 46-48 - need referencing for this and it has not been stated to date that EVOO is oleic acid rich.

line 50 - EVOO is not high in PUFA by comparison with other n-6 oils. Careful how you use the word rich to describe concentrations of fatty acids in oils. At the moment you are using it indiscriminately.

lines 65- 70  some of the comments about fish oil preventing heart attacks is not accurate given the current literature - the authors need to provide a more balanced view on this given all the MAs that are published on fish oil and MI. How is dietary EPA and DHA supplementation different to fish oil dietary supplements - please be specific.

line 83- i do not understand this last sentence

The introduction does not provide enough of a strong rationale for why they are reviewing this interesting area looking at the gut-liver axis. More background is needed on the gut-liver axis.

line 96 - not sure what 'successive' means here

Author Response

We would like to express our sincere gratitude to the Reviewer. We have made every attempt to fully address his/her comments in the revised manuscript and we feel that the quality of the new version of our document is substantially improved. The following paragraphs summarize our responses to the Reviewer’s comments.

Introduction

line 42 - the correct reference for 18;3n-3 is α-linolenic acid, not linolenic acid

We apologize for the mistake. We corrected the reference accordingly to Reviewer’s suggestion.

line 44- need to be more specific here, it is the inclusion of the 2 essential fatty acids that are important - linoleic acid and α-linolenic acid. No other fats have been shown to have essentiality - this is different from fats being linked to incidence of disease as you are describing here. Please rectify.

We apologize for the lack of clarity. We thank the Reviewer for the comment that led us to rewrite this section of the manuscript.

lines 46-48 - need referencing for this and it has not been stated to date that EVOO is oleic acid rich.

We apologize for the mistake. We modified the text in accordance with the Reviewer’s comment.

line 50 - EVOO is not high in PUFA by comparison with other n-6 oils. Careful how you use the word rich to describe concentrations of fatty acids in oils. At the moment you are using it indiscriminately.

We apologize for the inconvenience. We rewrote this section of the document.

lines 65- 70 some of the comments about fish oil preventing heart attacks is not accurate given the current literature - the authors need to provide a more balanced view on this given all the MAs that are published on fish oil and MI. How is dietary EPA and DHA supplementation different to fish oil dietary supplements - please be specific.

We thank the Reviewer to highlight this point and we apologize for the incompleteness of the reference proposed. We now incorporated a new section discussing the principal clinical trials on fish oil and the contradictory results postulated.

line 83- i do not understand this last sentence

We are grateful to the Reviewer for pointing out this poor clear sentence. We are now providing a modified version.

The introduction does not provide enough of a strong rationale for why they are reviewing this interesting area looking at the gut-liver axis. More background is needed on the gut-liver axis.

We thank Reviewer 2 for the constructive comment which helped us to improve the quality of our review. In the new version of the manuscript we have included more information on the gut-liver axis in the introduction.

line 96 - not sure what 'successive' means here

We apologize for the lack of clarity. We modified the text in order to improve our readability.

Reviewer 3 Report

The paper substantially is performing all requirements which are being put for examinations of this type. The review paper is well written and in general well organized. The presentation of individual chapters is logical and complete. All figures are graphically very readable. Short final conclusions are also very useful. References used in the article were correctly selected to presented problems.

Despite the very good language of article, I suggest the author re-reviewing the text by a native speaker.

Author Response

The paper substantially is performing all requirements which are being put for examinations of this type. The review paper is well written and in general well organized. The presentation of individual chapters is logical and complete. All figures are graphically very readable. Short final conclusions are also very useful. References used in the article were correctly selected to presented problems.

Despite the very good language of article, I suggest the author re-reviewing the text by a nativespeaker.

We express our sincere gratitude to the Reviewer for the careful and considered comments. We believe that the revision has resulted in a significantly improved manuscript.

Round 2

Reviewer 1 Report

In this version of the manuscript, the authors have correctly adressed all the concerns from this refeeree. Thus, i believe that the improvement of the manuscript placed it now to be published in the present form. No further observations to apply. 

Author Response

We would like to thank again Reviwer . His/her insightful comments and suggestions led us to obtain a improved version of the manuscript.

Reviewer 2 Report

Most of the changes are still impacted by the poor English. This continues to limit the readability and the interpretation of what has been written.

Line 54 - have not included EPA and DHA as n-3 fatty acids

line 56- these fatty acids are in the diet...they don't need to be supplemented

Line 58 - fats and oils are the same scientifically

line 59: lack of fatty acids does not cause depression, psoriasis etc... IT increases the risk. This seems to suggest poor understanding or perhaps interpretation of basic scientific concepts.

line 61- not sure why in primis has been included...all of oils have been studied.

lines  90-95 - need to include data from MA in this

Author Response

Most of the changes are still impacted by the poor English. This continues to limit the readability and the interpretation of what has been written.

We would like to thank Reviewer 2 for the effort and time he/she put in revising our manuscript. We have revised our paper along the lines outlined, and we honestly feel that his/her insightful suggestions helped us in improving the manuscript. The English has been now reviewed by a naïve-speaker.

Line 54 - have not included EPA and DHA as n-3 fatty acids

We apologize for our mistake. We now added also EPA and DHA as suggested.

line 56- these fatty acids are in the diet...they don't need to be supplemented

We apologize for the lack of clarity in this sentence, and we appreciate the Reviewer’s suggestion that led us to rewrite this section of the manuscript.

Line 58 - fats and oils are the same scientifically

We apologize for this inconvenience. We modified the text following Reviewer’s indication.

line 59: lack of fatty acids does not cause depression, psoriasis etc... IT increases the risk. This seems to suggest poor understanding or perhaps interpretation of basic scientific concepts.

We thank the Reviewer for highlighting this point. We changed the manuscript as suggested.

line 61- not sure why in primis has been included...all of oils have been studied.

We apologize for this inconvenience. We re-wrote this section of the manuscript.

lines 90-95 - need to include data from MA in this

We are grateful for the Reviewer’s careful comments. In the new version of the manuscript, we added data from meta-analysis regarding fish oil and cardiovascular diseases. If the Reviewer believes that the data reported in this topic are still incomplete, we are looking forward his/her kind suggestion about specific studies/papers to add.